# Structural Quality of Services and Use of Family Planning Services in Primary Health Care Facilities in Ethiopia. How Do Public and Private Facilities Compare?

**DOI:** 10.3390/ijerph17124201

**Published:** 2020-06-12

**Authors:** Gizachew Assefa Tessema, Mohammad Afzal Mahmood, Judith Streak Gomersall, Yibeltal Assefa, Theodros Getachew Zemedu, Mengistu Kifle, Caroline O. Laurence

**Affiliations:** 1School of Public Health, The University of Adelaide, Adelaide, SA 5005, Australia; afzal.mahmood@adelaide.edu.au (M.A.M.); judith.gomersall@adelaide.edu.au (J.S.G.); caroline.laurence@adelaide.edu.au (C.O.L.); 2Department of Reproductive Health, Institute of Public Health, University of Gondar, Gondar 196, Ethiopia; 3School of Public Health, Curtin University, Perth, WA 6201, Australia; 4South Australian Health and Medical Research Institute, Adelaide, SA 5000, Australia; 5School of Public Health, The University of Queensland, Brisbane, QLD 4072, Australia; yibeltalassefa343@gmail.com; 6Health System and Reproductive Health Research Directorate, Ethiopian Public Health Institute, Addis Ababa 1242, Ethiopia; tedi.getachew@yahoo.com; 7Institute of Public Health, College of Medicine and Health Sciences, University of Gondar, Gondar 196, Ethiopia; 8Federal Ministry of Health, Addis Ababa, 1234, Ethiopia; kiflemengistu@yahoo.com

**Keywords:** quality of services, family planning, public–private partnership, primary health care, Ethiopia

## Abstract

Background: Family planning (FP) is among the important interventions that reduce maternal mortality. Poor quality FP service is associated with lower services utilisation, in turn undermining the efforts to address maternal mortality. There is currently little research on the quality of FP services in the private sector in Ethiopia, and how it compares to FP services in public facilities. Methods: A secondary data analysis of two national surveys, Ethiopia Services Provision Assessment Plus Survey 2014 and Ethiopian Demographic and Health Survey 2016, was conducted. Data from 1094 (139 private, 955 public) health facilities were analysed. In total, 3696 women were included in the comparison of users’ characteristics. Logistic regression was conducted. Facility type (public vs. private) was the key exposure of interest. Results: The private facilities were less likely to have implants (Adjusted Odds Ratio (AOR) = 0.06; 95% Confidence Interval (CI): 0.03, 0.12), trained FP providers (AOR = 0.23; 95% CI: 0.14, 0.41) and FP guidelines/protocols (AOR = 0.33; 95% CI: 0.19, 0.54) than public facilities but were more likely to have functional cell phones (AOR = 8.20; 95% CI: 4.95, 13.59) and water supply (AOR = 3.37; 95% CI: 1.72, 6.59). Conclusion: This study highlights the need for strengthening both private and public facilities for public–private partnerships to contribute to increased FP use and better health outcomes.

## 1. Background

Family planning (FP) plays an important role in poverty reduction, women’s empowerment, human development, and maternal mortality reduction [1,2,3]. The 2019 Ethiopian Mini Demographic and Health Survey (EMDHS) reported that only 41% of married women were using contraceptive methods [4]. Despite improvements in contraceptive use in the last two decades [5,6,7], a previously conducted national survey indicated that a remarkable proportion (22%) of married women who wanted to delay or stop childbearing were not using any contraceptive methods [4]. It was also noted that within the first 12 months of married or in-sexual union women’s contraceptive use, as many as 35% discontinue contraceptive use [7]. Research has shown that contraceptive discontinuation could be due to the poor quality of care in FP services reflected by the lack of information provision about the side effects of the methods they received, lack of privacy during counselling and long waiting times [8,9,10]. Improving quality of care in FP services can help maintain contraceptive use for new FP users and help generate demand by non-users [11,12].

In Ethiopia, FP services are provided by all levels of the health system [13], with Primary Health Care Unit (PHCU) facilities being the major source for FP methods for women. In terms of the type of facilities where women source FP services, one in seven (14%) of them have accessed FP methods from private facilities [7].

While the private sector provides a small proportion of FP services in Ethiopia, recent health and FP policies and strategies have recognised the potential key role of the private health sector in improving access for FP services [14,15,16]. A report in 2015 showed that Public-Private Partnerships (PPP) in Ethiopia, established between government and private-for-profit and private-non-profit organisations in the provision of primary health care services, have improved access to quality health services including FP [17]. As a result, there is an effort to strengthen the engagement of private facilities in the provision of FP services in Ethiopia [14]. Moreover, private facilities are targeted to be part of the quality improvement programmes. The Ethiopian National Health Care Quality Strategy (2016–2020), which aims to transform the quality of health care in Ethiopia, has indicated that the private sector needs to be included in the process of quality assurance and evaluation of quality improvement [18].

A study by the Lancet Global High Commission urges the necessity of quality of care services as a basic human right and cited quality as the central element of health care system [19]. However, it was found that one in three people in low- and middle-income countries (LMICs) had negative experiences with their health system in the areas of attention, respect, communication, and length of visit. The quality of care in FP services can be measured in a number of ways, but the Donabedian and the Bruce–Jain frameworks are deemed suitable for measuring quality of care in FP services and being widely implemented since 1990 [20,21]. The Donabedian model conceptualises quality of care as a linear model comprising three components—structure, process, and outcome. The structure component of quality of care includes the attributes of material resources such as facilities, equipment, and commodities; human resources such as the availability of adequate and trained staff; and organisational structure such as reward systems and quality assurance systems. The process dimension focuses on the way the health care services are delivered and includes ‘provider-client interaction’. The last component of quality of care is the outcome obtained from provider and client interaction in the FP facilities, such as client’s satisfaction, changes in knowledge, and other subsequent long-term aspects such as a reduction in fertility and mortality. These three components are interlinked, as good structure increases the likelihood of good process, and good process increases the likelihood of a good outcome [21,22]. The Bruce–Jain Framework, underpinned by the works of the Donabedian, identified six key elements of quality of care—choice of methods, information given to clients, technical competence of providers, interpersonal relations, follow-up mechanisms, and appropriate constellation of services [20]. Previously, the authors have highlighted the relationship between the outcome aspects of quality of care and structure and process aspects of quality of care in FP services in a preceding study, although the analysis did not distinguish between type of facility—public or private [23]. The present study is, therefore, particularly focusing on the structural quality of FP services. Consequently, we employed the Donabedian framework as it gives a proper assessment of the facility’s structural quality of services in terms of human resource and material infrastructure, which are the basic precursors for good quality of care in FP services.

Despite only a relatively small proportion of contraceptive users accessing services from private facilities in Ethiopia [7], these facilities are popular sources of FP methods in several LMICs, contributing up to 40% of the share in Bangladesh and Cambodia and as many as 50% of the sources of FP methods in Nigeria [24]. There is currently little evidence on the quality and scope of FP services provided in private facilities in Ethiopia or how they compare to the services provided in public facilities. This is important, particularly in a climate where the government is using the public–private partnership to improve the provision of FP services in the country. Studies based in Kenya [25], Pakistan [26], and Congo [27] have shown that private facilities have a better quality of FP services in terms of infrastructure and the availability of trained providers. On the other hand, studies done in Indonesia [28] and Jamaica [29] have shown that public facilities provided a better quality of care for antenatal and FP clients, respectively. Overall, the limited available evidence on quality of care differences between public and private facilities indicates that differences are likely to vary by the country context. This study aims to compare the ‘structural’ aspects of quality of care in FP services between public and private Primary Health Care Unit (PHCU) health facilities in Ethiopia. In addition, this study investigates the characteristics of women who accessed FP services in private and public health facilities.

## 2. Methods

### 2.1. Design and Setting

This study is based on secondary data analysis of two nationally representative surveys; the Ethiopian Services Provision Assessment Plus (ESPA+) data, facility-based data collected in 2014 and the Ethiopian Demographic and Health Survey (EDHS) data, community-based data collected in 2016.

Ethiopia, with nearly 108 million people and an average total fertility rate of 4.6 children per women, is the second most populous country in Africa, next to Nigeria [7,30]. Most of the people receive health services in PHCU facilities and these facilities provide a wide range of basic health services, including FP. Secondary and tertiary care is provided at general hospitals and specialised hospitals [14]. While FP services are provided in both PHCU and hospitals, most (96%) women receive FP services at PHCU facilities [7].

### 2.2. Data Sources and Sample

This study used two data sources. The dataset used for the comparison of the structural quality of services was the ESPA+ 2014 survey data. The present study considered data obtained from the facility inventory survey, which involved a total of 1,327 health facilities. We excluded those facilities that reported not being functional during the survey (*n* = 145), not providing FP services (*n* = 64), and general and specialised/referral hospitals (*n* = 24). Consequently, we included a weighted sample of 1094 primary health care (private = 139, public = 955) facilities. The ESPA+ survey used standardised tools, data collection methods, and quality assurance strategies. Details about the ESPA+ survey are published elsewhere [31].

To compare the characteristics of women who accessed FP services in public versus private PHCU facilities, we used data from the EDHS 2016. The EDHS is a cross-sectional survey conducted every five years to measure the demographic and health-related characteristics of the population [32]. The EDHS 2016 survey interviewed 15,683 women of reproductive age. A stratified, two-stage cluster sampling procedure was used to include a nationally representative sample of reproductive aged women. Of the 3,884 women who accessed modern contraceptive methods during the survey, 3,696 sourced contraceptive methods from the PHCU facilities and were included in the final analysis. Details of the sampling, methods and design of the EDHS 2016 are published in a report elsewhere [7].

### 2.3. Variables

The variables used for comparing structural factors impacting on the quality of services in public and private PHCU facilities were based on the facility inventory assessment of the ESPA+ 2014 survey [31].

We included structural variables that reflected the material structure, such as the facility’s infrastructure (basic amenities), the availability of equipment and supplies; human resources such as health provider availability and trained provider availability; and organisational structure such as quality assurance system and supervision in the past six months, the availability of FP guidelines/protocols, the availability of other maternal and child health services such as the presence of antenatal, delivery, and the availability of range of modern contraceptive methods. Detailed descriptions of the variables are provided in Appendix A.

For the analysis of the characteristics of women who accessed FP services from the two different types of PHCU facilities, we included variables related to the sociodemographic characteristics and exposure to FP media from the 2016 EDHS [7] (see Appendix A).

### 2.4. Data Analysis

Descriptive and summary statistics were used to describe the characteristics of the facilities in ESPA+ 2014 survey and women who accessed FP services in EDHS 2016. The analysis was conducted after applying weighting of samples.

Bivariate and multivariate logistic regression analyses were used to compare structural variables between public and private PHCU facilities. Odds Ratios (OR) and 95% Confidence Intervals (CI) were calculated to determine the association and level of significance. Variables with *p*-value of less than 0.2 in the bivariate logistic regression analysis were adjusted for facility location, as this was considered to be the main confounding variable [27]. While taking the structural variables as an outcome for comparison, the facility type (private vs. public) was used as the key exposure of interest. We also conducted bivariate and multivariate logistic regression analyses to examine the association between women’s characteristics and the facility type they used for FP services. In this analysis, women’s characteristics were considered as independent variables; facility type (private vs. public) was the outcome variable. STATA 14 (StataCorp, College Station, Texas USA) was employed for the analysis.

### 2.5. Ethical Consideration

Ethical approvals were obtained from the Scientific and Ethical Review Committee (SERC) at the Ethiopian Public Health Institute (EPHI) (EPHI 6.13/966) and ethics exemption from was received from the Human Research Ethics Committee (HREC) at the University of Adelaide (App. No: 0000021084). Permission to use the publicly available EDHS data was granted from DHS program.

## 3. Results

### 3.1. Description of Facility’s Location, Structure, and Provisions of Reproductive and Child Health Services in PHCU Facilities

In the included PHCU facilities, 16% (22) of private and 22% (206) of public facilities reported that there were health providers available twenty-four hours/seven days in those facilities. While 72% (99) of the total private health facilities comprised lower clinics, 81% (773) of the total public facilities comprised health posts. The median number of days in a week that FP services were provided was 6.3 days/week in private facilities and 5.2 days/week in public facilities. When asked to show the FP guidelines/protocols utilised in the facility for guiding the provision of FP services, only 29% of the private and nearly half (49%) of public facilities were able to do this. While the majority (92%) of private facilities charged a ‘user fee’ for FP services, nearly all (99%) of public facilities reported that FP services were provided for free. Diagnosis and treatment for Sexually Transmitted Infections (STI) were provided in 35% and 90% of public and private facilities, respectively. Injectable (95.5–98.7%) and male condoms (87.4–87.7%) were the most commonly available methods in both facilities (Figure 1, Appendix A).

### 3.2. Structural Quality of FP Services in Public and Private PHCU Facilities

Private facilities were less likely to have IUD (AOR = 0.22; 95% CI: 0.13, 0.38) and implants (AOR = 0.06; 95% CI: 0.03, 0.12), but were more likely than public facilities to have emergency contraceptive methods (AOR = 3.81; 95% CI: 2.37, 6.10). When comparing services’ provision environment factors, private facilities were less likely to have healthcare provider available 7 days a week in the facility (AOR = 0.35; 95% CI: 0.18, 0.69), healthcare providers who received FP training in the past 24 months (AOR = 0.23; 95% CI: 0.14, 0.41), or FP guidelines/protocols (AOR = 0.33; 95% CI: 0.19, 0.54) than the public facilities (Table 1).

Comparing the availability of basic amenities (infrastructure), private facilities were more likely than public facilities to have functional cell phones (AOR = 8.20; 95% CI: 4.95, 13.59) and water supply (AOR = 3.37; 95% CI: 1.72, 6.59). When comparing the availability of equipment, private facilities were less likely than public facilities to have pelvic model for Intrauterine Device (IUD) demonstration (AOR = 0.39; 95% CI: 0.21, 0.76), and penile model for condom demonstration (AOR = 0.40; 95% CI: 0.21, 0.76). However, the availability of stethoscope (AOR = 7.88; 95% CI: 3.49, 17.73), examination light (AOR = 8.19; 95% CI: 4.86, 13.79), and examination couch (AOR = 14.11; 95% CI: 5.84, 34.08) were higher in private facilities than public facilities.

It was found that private facilities were less likely to have antenatal care (AOR = 0.05; 95% CI: 0.02, 0.10) and normal delivery services (AOR = 0.15; 95% CI: 0.08, 0.30). However, the diagnosis and treatment of STI (AOR = 8.51; 95% CI: 4.64, 15.61) were more likely to be provided in private facilities than public facilities (Table 1).

### 3.3. Association of Women’s Characteristics with Facility Type Where They Accessed FP Services

Of the 3696 women who accessed FP methods, 3110 (84%) did so at public facilities and 586 (16%) from private health facilities. While the majority (82%) of women in rural areas accessed FP services from public facilities, half (50%) of them accessed FP services in private facilities (Table 2).

The women who accessed FP services from private facilities were significantly more likely to reside in an urban area (AOR = 3.91; 95% CI: 1.71, 4.95), be Muslim (AOR = 1.63; 95% CI: 1.07, 2.48), and to be employed/working (AOR = 1.31; 95% CI: 1.01, 1.96). When compared to women with no children, women having 1–2 children (AOR = 0.27; 95% CI: 0.15, 0.47), 3–4 children (AOR = 0.23; 95% CI: 0.11, 0.46), and 5 or more children (AOR = 0.18; 95% CI: 0.08, 0.41) were 73%, 77%, and 82% less likely to access FP services in private facilities, respectively. The analysis showed no association between the source of FP methods and the women’s current marital status, wealth status, and their decision-making power in the household (Table 3).

## 4. Discussion

This study found that there were differences in the structural factors that may impact the quality of FP services between private and public health facilities. It also demonstrated that the characteristics of women who accessed FP services from private facilities were different from those who accessed these services from public facilities.

A previous study found in private facilities that the presence of trained providers at all times was the most important structural factor in terms of access to services for clients, irrespective of their income status [33]. This study showed that FP services’ environment factors, including the availability of training for the FP provider, seven-days availability of health providers’, and the provision of FP guidelines, was less available in the private facilities when compared to their availability in public health facilities. This finding suggests the need for the government to improve the FP services environment by increasing the participation of private facilities during the provision of FP training and distribution of FP guidelines. While the contraceptive methods mix between short- and long-acting methods needs to be balanced, gaps were identified in the availability of long-acting methods, which may partly explain why the vast majority (66%) of women in Ethiopia rely on a single contraceptive method [7].

The private health facilities were found to have better basic infrastructure such as functional cell-phones and water supply than public facilities, a finding consistent with a similar study conducted in Kenya [25]. This result reflects the urban location of most private facilities, with urban areas having better access to mobile-network and water sources [34]. This study showed some mixed findings vis-à-vis the availability of equipment and FP methods in the provisions of FP services of public and private facilities. While the private facilities were better in terms of having basic equipment that is also needed for services beyond FP, such as stethoscopes, examination lights, and examination couches, the availability of FP-specific equipment such as a penile model and Intrauterine Device (IUD) model for male condoms and IUD methods insertion demonstrations were less available in public facilities. This difference was also reflected in the available FP methods in that commonly utilised FP methods, which included IUDs and implants, were less likely to be provided in private facilities than public facilities in Ethiopia. These findings may arise from a lack of policies that facilitate the supply of essential commodities for the provision of FP services in private facilities [35]. Despite the more limited ranges of FP services in the private facilities, it was found that STI services and emergency contraception are more likely to be available in the private facilities than the public facilities. A study conducted in Nigeria and Kenya supports our finding, in that emergency contraceptive methods were mostly available in private health facilities [36].

Equipment such as stethoscopes and examination couches are used not only for FP but for most other health problems with which the patients present, and that is probably the reason that this equipment is better sourced and made available in private facilities.

The profile of the clients using private facilities for FP services was not surprising. The private facilities attract employed or working women and women with few or no children which could be because they are mostly found in urban areas where women have lower fertility than their rural counterparts. In addition, the provision of FP services for greater number of days in private facilities than public facilities (6.3 days per week versus 5.1 days per week in public facilities) may also account for the reason certain women use these facilities. For example, the more flexible opening hours would allow employed women access to FP services that may not be accommodated in the shorter opening hours found in the public facilities. This implies that the public facilities located in rural areas need to expand the availability of FP services to cater to the needs of women living in these areas. This finding is consistent with a study conducted in 57 LMIC, that found that women residing in urban areas used private sectors more than women in rural areas [37].

The study has a number of implications. If the private facilities are to play an important role in the provision of quality FP services in Ethiopia, there needs to be introduction of policies that support expanding their client base so that the private facilities attract not just urban residents, employed/working women, and women with no or few children. In order for a broader range of women to access privately provided FP services, this may require subsiding the costs of FP service in the private facilities. With the expansion of the public–private partnerships in the provision of health care services in Ethiopia, it is also important to ensure a high quality of services is provided in both private and public sectors. This study showed that there is a need for improvement with regard to some important structural quality aspects of the FP services in private facilities. The application of government policies and guidelines on quality FP services to the private sector would contribute to improving the structural quality of services. There are also lessons to be learnt from the private facilities that can be applied to public facilities. For example, the government should work on improving basic infrastructure and the availability of equipment in the public facilities and to strengthen public facilities to be able to cater to the needs of women with different backgrounds.

This study has the following strengths. The study uses nationally representative datasets collected using standardised methodologies and instruments. It was also the first large study that compared the quality of services between public and private facilities. However, the following important limitations should be considered. The analysis of quality of care in FP services was limited to analysis of the structural aspects and did not include process and outcome aspects. This limitation arose because the study used data collected in a national survey, where the number of FP clients in private sectors from whom data related to process and outcome variables was collected was very small. The inclusion of process (e.g., waiting time, provision of information) and outcome variables (e.g., client satisfaction) in future research would help to achieve a more comprehensive understanding of the state of quality of FP services. However, an understanding of the structural aspect does provide good insight which could be used to strengthen FP services. Another limitation was that, while the EDHS 2016 survey collected data for those women who accessed FP services from primary, general, and specialised hospitals, in the dataset these were combined into one variable. As a result, the types of facilities included in the ESPA+ 2014 and the EDHS 2016 were not matched. Despite primary hospitals being categorised as part of the PHCU facilities in Ethiopia, given that the EDHS 2016 survey put the data for contraceptive users at hospital level together, we were not able to distinguish women who specifically accessed FP services from the primary hospitals only, and therefore data related to the characteristics of women who accessed FP services in primary hospitals were not included. However, since only 4% of women were accessed FP services from hospitals [7], we expected that the number of women who accessed FP services from primary hospitals would be low. The ESPA+ data were not linked to the EDHS data. However, as both surveys involved data collection in all 11 administration regions in Ethiopia, it could be assumed that women included in the EDHS household survey are likely to access FP services from the health facilities that were included in the ESPA+ facility-based survey.

In conclusion, it was found that there were differences in the structural quality of FP services between public and private PHCU facilities. When compared to the public facilities, private facilities were deficient in terms of the availability of supplies, FP methods, trained providers, FP guidelines, and quality assurance activities, but were better in terms of some of the basic infrastructure and equipment. Women who accessed FP services from private facilities were different from those who accessed services from public facilities. This study alerts the need for strengthening both private and public facilities for public–private partnerships to contribute to increased FP use and better health outcomes. While the government needs to support public health facilities in improving the availability of family planning methods to improve methods mix, it is also necessary to engage the private facilities during family planning guidelines distributions, and for their healthcare providers to get the opportunity for family planning training.

## Figures and Tables

**Figure 1 ijerph-17-04201-f001:**
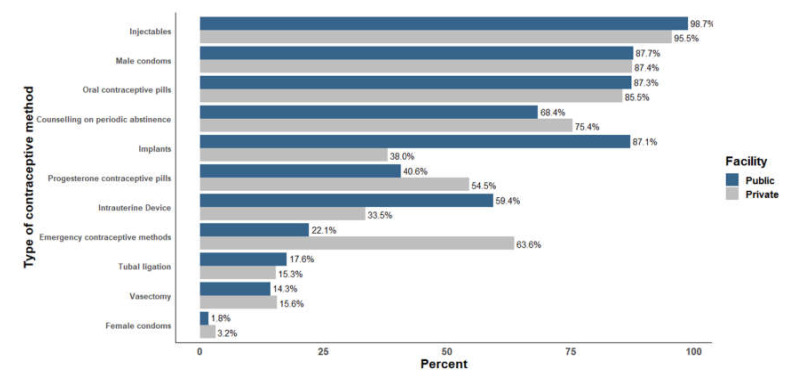
Percentage of public and private PHCU facilities with available contraceptive method, Ethiopian Services Provision Assessment Plus (ESPA+) 2014.

**Table 1 ijerph-17-04201-t001:** Unadjusted and adjusted regression model for comparing structural quality of services in private vs. public Primary Health Care Unit (PHCU) facilities in Ethiopia, ESPA+ 2014 (*n* = 1094).

Variables	OR ^1^ (95% CI)	
	COR ^1^ (95% CI)	AOR (95% CI) ^1,2^
**Availability of contraceptive methods**		
Availability of progesterone contraceptive pills	1.73 (1.15, 2.60) *	1.31 (0.83, 2.09)
Availability of IUD	0.35 (0.22, 0.54) *	0.22 (0.13, 0.38) *
Availability of implants	0.10 (0.06, 0.16) *	0.06 (0.03, 0.12) *
Availability of emergency contraceptive methods	6.17 (3.98, 9.59) *	3.81 (2.37, 6.10) *
**Services provision environment**		
Health provider availability of twenty-four hours/seven days	0.68 (0.42, 1.11) *	0.35 (0.18, 0.69) *
Trained provider availability	0.28 (0.19, 0.43) *	0.23 (0.14,0.41) *
Quality assurance system	0.10 (0.79, 1.79) *	0.07 (0.02, 0.21) *
FP guidelines/protocols	0.43 (0.04, 0.28) *	0.33 (0.19, 0.54) *
Client chart/record to document the client’s clinical data	0.23 (0.16, 0.37) *	0.22 (0.13, 0.36) *
Supervision in the past six months	0.75 (0.50, 1.15) *	0.91 (0.57, 1.47)
Private room for providing counselling services	1.16 (0.89, 3.18) *	0.88 (0.46, 1.70)
**Facility’s basic infrastructure**		
Availability of functional landline telephone	3.43 (2.08, 5.69) *	0.64 (0.30, 1.37)
Availability of functional cell phone	9.18 (5.56, 15.15) *	8.20 (4.95, 13.59) *
Access to email at least two hours on a day	6.22 (2.68, 14.48) *	2.01 (0.78, 5.50)
Availability of functional computer	1.20 (0.75, 1.92)	0.18 (0.07, 1.44)
Access to water supply	5.25 (2.89, 9.57) *	3.37 (1.72, 6.59) *
Availability electricity supply/generator	1.52 (1.02, 2.26) *	1.11 (0.69, 1.76)
**FP services equipment**		
Availability of stethoscope	9.17 (4.28, 19.61) *	7.88 (3.49, 17.73) *
Availability of examination light	8.1 (5.04, 13.07) *	8.19 (4.86, 13.79) *
Availability of exam couch	19.01 (8.27, 43.72) *	14.11 (5.84, 34.08) *
Availability of sample FP methods	1.44 (0.95, 2.17)	0.87 (0.55, 1.37)
Pelvic model for demonstrating IUD use demonstration	0.56 (0.31, 1.01) *	0.39 (0.21, 0.76) *
Model for demonstrating condom use	1.03 (0.61, 1.71) *	0.40 (0.21, 0.76) *
**Availability of additional maternal and child health services**		
Antenatal care services	0.06 (0.03,0.13) *	0.05 (0.02, 0.10) *
Normal delivery services	0.28 (0.18,0.43) *	0.15 (0.08, 0.30) *
Under-five health services	0.18 (0.06, 0.56) *	0.15 (0.05, 0.45) *
Services for the prevention of mother-to-child transmission of HIV	0.48 (0.30, 0.77) *	0.15 (0.08, 0.32) *
Diagnosis and treat for STI	13.5 (7.70, 23.8) *	8.51 (4.64, 15.61) *

^1^ Public facility was taken as a reference in the analysis. ^2^ The final model was adjusted for facility location. * *p*-value < 0.05. FP—Family Planning, IUD—Intrauterine Device, STI—Sexual Transmitted Infections, HIV—Human Immunodeficiency Virus, OR—Odds Ratio COR—Crude Odds Ratio, AOR—Adjusted Odds Ratio.

**Table 2 ijerph-17-04201-t002:** Characteristics of women accessing FP services from public and private health facilities, EDHS 2016 (*n* = 3696).

Women Characteristics	Private (*n* = 586)	Public (*n* = 3110)
	Frequency (%)	Frequency (%)
**Age in Years**		
15–24	200 (34.1)	737 (23.7)
25–34	278 (47.5)	1467 (47.2)
35+	108 (18.4)	906 (29.1)
**Marital Status**		
Currently married/in union	58 (9.9)	222 (7.1)
Not currently married	528 (90.1)	2888 (92.9)
**Place of Residence**		
Urban	291 (49.4)	561 (18.0)
Rural	297 (50.6)	2549 (82.0)
**Religion**		
Orthodox	336 (57.4)	1603 (51.6)
Muslim	134 (22.9)	629 (20.2)
Protestant	109 (18.6)	833 (26.8)
Other/missing	7 (1.1)	45 (1.4)
**Highest Educational Status**		
None	196 (33.4)	1773 (57.0)
Primary	213 (36.5)	969 (31.1)
Secondary+	177 (30.1)	368 (11.8)
**Partner’s Educational Status( *n* = 3416) ^$^**		
None	125 (23.6)	1237 (42.8)
Primary	216 (41.0)	1161 (40.2)
Secondary+	187 (35.4)	490 (7.0)
**Working/Occupational Status**		
Not working	206 (35.1)	1445 (46.5)
Working/employed *	380 (64.9)	1664 (53.5)
**Wealth Index**		
Poor	85 (14.6)	990 (31.8)
Middle	89 (15.1)	679 (21.8)
Rich	412 (70.3)	1440 (46.3)
**Number of Living Children**		
0	142 (24.1)	241 (7.8)
1–2	231 (39.4)	1120 (36.0)
3–4	142 (24.1)	906 (29.1)
5+	72 (12.3)	842 (27.1)
**Exposure to FP Media**		
No	293 (50.0)	2174 (69.9)
Yes	293 (50.0)	936 (30.1)
**Decision Making Power in the Household**		
No	176 (30.2)	1021 (32.8)
Yes	410 (69.8)	2089 (67.2)

FP—Family Planning ^$^ Sample was taken for clients who are currently married * Working includes those women who describe themselves as employed or engaged in a work that paid them in cash or in kind.

**Table 3 ijerph-17-04201-t003:** Association of women’s characteristics with the types of health facility where they accessed FP services (private vs. public), EDHS 2016 (*n* = 3696).

Women Characteristics	OR^1^ (95% CI)	
	COR ^1^ (95% CI)	AOR ^1^ (95% CI)
**Age in Years**		
15–24	1	1
25–34	0.69 (0.51, 0.96) **	1.02 (0.65, 1.64)
35+	0.44 (0.29,0.56) ***	1.07 (0.56, 2.06)
**Place of Residence**		
Urban	4.44 (3.05, 6.46) ***	3.91 (1.71, 4.95) ***
Rural	1	1
**Religion**		
Orthodox	1	1
Muslim	1.02 (0.67, 1.53)	1.63 (1.07, 2.48) *
Protestant	0.62 (0.41, 0.96) *	0.93 (0.56, 1.53)
Other/missing	0.70 (0.16, 3.02)	1.24 (0.25, 6.28)
**Highest Educational Status**		
None	1	1
Primary	1.99 (1.41, 2.83) ***	1.8 (0.71, 1.65)
Secondary+	4.34 (2.97, 6.33) ***	0.91 (0.52, 1.60)
**Partner’s Educational Status (*n* = 3416) ^$^**		
None	1	1
Primary	1.85 (1.26, 2.72) **	1.49 (0.99, 2.23)
Secondary+	3.79 (2.44, 5.87) ***	1.64 (1.01, 2.70) *
**Working/Occupational Status**		
Not working	1	1
Working ^$^	1.60 (1.18, 2.17) **	1.35 (1.01, 1.96) *
**Wealth Index**		
Poor	1	1
Middle	1.51 (0.97, 2.37) ***	1.54 (0.96, 2.47)
Rich	3.31 (2.21, 4.96) ***	1.51 (0.90, 2.54)
**Number of Living Children**		
0	1	1
1–2	0.35 (0.22, 0.55) ***	0.27 (0.15, 0.47) **
3–4	0.27 (0.17, 0.42) ***	0.23 (0.11, 0.46) **
5+	0.14 (0.08, 0.26) ***	0.18 (0.08, 0.41) **
**Exposure to FP Media**		
No	1	1
Yes	2.33 (1.70, 3.19) ***	0.97 (0.66, 1.44)

FP—Family Planning OR—Odds Ratio COR—Crude Odds Ratio AOR—Adjusted Odds Ratio, ^1^ Public facility was taken as a reference in the regression analysis. ^$^ Working includes those women who describe themselves as employed or engaged in a work that paid them in cash or in kind *** *p*-value < 0.001 ** *p*-value < 0.01 * *p*-value < 0.05.

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
