# Peer review of "Structural Quality of Services and Use of Family Planning Services in Primary Health Care Facilities in Ethiopia. How Do Public and Private Facilities Compare?"

_ijerph, 2020, doi:10.3390/ijerph17124201_

Round 1
Reviewer 1 Report
This paper is presenting on an important topic - of examining the differences between private and public family planning clinics in terms of the clinics and who attends them. Some of my overarching comments are that the rationale for government investment in private facilities needs to be made more clear - since the data show that the public facilities are operating at a subpar level there isn't much to be said for splitting the few resources that the government is providing to even more clinics. It is also worrisome that the wealthier attend the private clinics - which would mean a diversion of resources from the poorest to those who are better off.
Introduction/Methods
Differing rates of public vs private health center use noted in paper in different spots- if 14% or 4%?
Results
Perhaps you are reporting on too many variables - in that the findings get a bit lost and the reader must look for the main findings in the tables. Highlight the main findings in the text and perhaps present on fewer findings, or find ways to group the individual indicators into group indicators, and in the text highlight the main findings of interest. To me, having methods available and the skills and resources necessary to provide them are the most important.
There are a number of places in the Table 1 were a decimal is missing in the percentage reported for public facilities.
Given that family planning is not a service that is needed 24 hours a day – this seems like an odd metric to use to compare private and public health facilities. Is it a goal to provide FP services 24 hours a day?
81% of public sites were public? Why not 100% were public?
The rural/urban difference deserves more attention – as it is quite stark.
Interesting differences by IUD and EC – also interesting to see that tubal ligation and vasectomy are offered at the same rate
It is odd that CIs are presented in the text but not in the table – they should be added to the table – then it would not be necessary to have two tables for these data. What are the ORs adjusted for – just urban/rural status or something more specific to geography?
Private facilities seem to cater to youth, unmarried - but as more expensive, having the EC and STI services at a greater rates seems to disadvantage poor youth from access
Odd that FP methods were not at 100% at public facilities - it strikes me that would be a necessary starting point - not sharing resources with private facilities
In general, the tables are quite long and organized in a way that doesn't help the reader find the main differences immediately. They could be more succinct and organized in a manner to draw attention to the main findings. For example, starting with facility type when the facility type differs for the two types of service provision doesn't actually make sense and should be removed from the table.
Also, how can a family planning facility not have pills and condoms. Is it possible that facilities were included in the sample that did not actually provide family planning services?
And if it is due to religious health facilities - how is the government involved in these facilities to make sure that citizens can still access family planning methods at or nearby these facilities?
Discussion
I would have loved to read more about how the government can attend to the gaps found in the public facilities - and then more on whether sharing the limited government resources with even more facilities, private in this case, would take away from the public facilities or not.
Author Response
Dear reviewer,
We have carefully reviewed the feedback on our paper entitled “Structural quality of services and use of family planning services in primary health care facilities in Ethiopia: How do public and private facilities compare?” We would like to acknowledge the reviewers for their thoughtful comments and suggestions that helped us to improve the quality of the manuscript.
Please find attached a point-by-point response for each of the comments from and the revised manuscript. We presented our responses in blue. We also highlighted the revisions we have made in a track-changes document. Thank you for your continued consideration of our manuscript.
Sincerely,
Dr Gizachew Tessema
On behalf of the authors

Reviewer 2 Report
The article presents a well-conceived study to respond to the research questions posed around a very important topic. It provides insight to the provision of FP services in Ethiopia which experiences a significant birthrate, maternal mortality and poverty. Utilizing the Donabedian Model for structural quality assessment is sound and aligns with current consideration of quality in healthcare organizations. The use of secondary data analysis has inherent limitations and thus the novel contribution of the study is modest.
- Significance: The significance lies with informing providers, communities and policymakers, perhaps grantors around the needs of the women in Ethiopia. The analysis is sound and appropriate. While the tables are long, the discussion of the data presented is often brief, particularly the demographic profile of the women which warrants greater consideration. The 'picture' of the women included is eye-opening. Some discussion of these as contributing factors to the outcome would add to the insight gained by including all of these factors. The title for Table 4 is confusing as is it does not seem to describe the data presented. Please review and clarify. The conclusions are supported by the data but are more descriptive than analytic. The implications for women receiving care from each type of facility is important as, interestingly, women receiving care from private facilities do not have the same access to IUDs as those receiving care in public facilities. As such, if one assumes private is better, it is an incorrect assumption in this regard)
- Quality of Presentation: The article follows a very acceptable format for a peer-reviewed journal. Some minor editing is needed. See Line 244, the list seems to include an item that does not belong there.See comment above for request to rethink the title for Table 4. Line 243 has a misuse of the word fort, Line 247 and 248 should be reviewed. For these very long, broken tables, repeating the header row would help the readability.
- Scientific Soundness: The student is well-designed including the methods and software. The analyses are appropriate. The authors clearly identify significant results. This study could be replicated with the description of the methods provided.
- Interest to the Readers: The paper is of interest to those committed to women's health, reproductive health and justice, and the quality of healthcare. Again, the study is more descriptive and does not yield very startling results. However, it does identify some of the shortcomings of the FP facilities and the associated system in which they exists. As well, the assumption (often) that private care is 'better' is not supported by the study result and this should be of interest to policymakers as well as providers.
Author Response
Subject: Revised version of manuscript No. ijerph-809849
Dear reviewer,
We have carefully reviewed the feedback on our paper entitled “Structural quality of services and use of family planning services in primary health care facilities in Ethiopia: How do public and private facilities compare?” We would like to acknowledge the reviewers for their thoughtful comments and suggestions that helped us to improve the quality of the manuscript.
Please find attached a point-by-point response for each of the comments from our reviewer and the revised manuscript. We presented our responses in blue. We also highlighted the revisions we have made in a track-changes document. Thank you for your continued consideration of our manuscript.
Sincerely,
Dr Gizachew Tessema
On behalf of the authors
